# Virtual Stars, Real Fans: Understanding the VTuber Ecosystem

## ABSTRACT

Livestreaming by VTubers — animated 2D/3D avatars controlled by real individuals — have recently garnered substantial global followings and achieved significant monetary success. Despite prior research highlighting the importance of realism in audience engagement, VTubers deliberately conceal their identities, cultivating dedicated fan communities through virtual personas. While previous studies underscore that building a core fan community is essential to a streamer's success, we lack an understanding of the characteristics and behaviors of viewers of this new type of streamer. Gaining a deeper insight into these viewers is critical for VTubers to enhance audience engagement, foster a more robust fan base, and attract a larger viewership. To address this gap, we conduct a comprehensive analysis of VTuber viewers on Bilibili, a leading livestreaming platform where nearly all VTubers in China stream. By compiling a first-of-its-kind dataset covering 2.7M livestreaming sessions, we investigate the characteristics, engagement patterns, and influence of VTuber viewers. Our research yields several valuable insights, which we then leverage to develop a tool to "recommend" future subscribers to VTubers. By reversing the typical approach of recommending streams to viewers, this tool assists VTubers in pinpointing potential future fans to pay more attention to, and thereby effectively growing their fan community.

## 1 INTRODUCTION

Live streaming has gained significant popularity globally, encompassing a wide range of themes, including personal experiences [26, 28, 42], artistic creation [8, 27], educational content [6, 29, 31], and gaming sessions [5, 13]. As part of this trend, we have recently witnessed the emergence of a novel type of streamer, referred to as *VTubers* (Virtual Youtubers) [30]. A VTuber is an animated 2D or 3D virtual avatar that performs in live video streams. These streams are produced using tools like Live2D [25] that capture the actor's movements and control the avatar accordingly. The avatar is therefore voiced and controlled by a specific person. VTuber content covers a wide range of topics, including anything that does not require a real human to present.

Although seemingly fringe, VTubers have been gaining significant popularity across various platforms, with dedicated fan bases and corporate sponsorship deals. There are tens of thousands of highly active and influential VTubers — the most popular ones have millions of followers [47]. Thanks to the various monetization mechanisms offered by live streaming platforms, VTubers earn substantial incomes. Among the top 50 highest-earning YouTubers based on all-time superchats (viewer donations), 31 are VTubers, each earning between $1.1 million and $3.2 million USD [37].

In contrast to prior studies showing that a sense of realism is a crucial factor in audience engagement [11, 41], VTubers intentionally conceal their real identities and present themselves to viewers through virtual avatars. This approach completely reconstructs the self-representation [3, 49] of the streamer, and also the relationship between the streamer and their fanbase [22, 30, 45, 52]. This is driven by an intersection of Japanese otaku culture and idol culture [7, 30], with the core viewer communities of VTubers exhibiting similar traits. As a result, these online viewer communities play a much more significant role compared to those of other streamers. The core viewers of a VTuber usually create and share their own subculture, and are active and passionate. Core viewers engage not only in live streaming but also contribute in various other ways to expand the VTuber's influence. For instance, there are over 420,000 dedicated fan artworks on Pixiv [36] for VTubers affiliated with Hololive (a "celebrity" VTuber agency).

Consequently, building such a core viewer community is essential to a VTuber's success, both in terms of fame and monetization. Yet, we still possess only a limited knowledge of the characteristics and behaviors of VTuber's viewers. We argue that gaining a deeper understanding is crucial for VTubers to enhance engagement with their audience, foster a stronger fan community, and thus attract more viewers (thereby generating greater monetary income). This insight can be especially beneficial for self-operated and startup VTubers, who do not belong to a management agency, as it will help them better understand their audience and identify potential core viewers.

To bridge this gap, we conduct the first large-scale measurement study of VTuber core viewers, focusing on Bilibili as an exemplar platform. Bilibili is a major website for user-generated video and live streaming. It currently hosts the largest number of VTubers in China, with almost all Chinese VTubers streaming on the platform. Importantly, on Bilibili, core viewers can be easily identified through a membership system, whereby users can pay a monthly fee to gain membership for a streamer. That is, "members" are the core viewers. This membership is similar to a paid subscription on Twitch but more expensive and with additional privileges *e.g.,* participation in a private chat group. Consequently, purchasing a membership signifies a strong commitment to the streamer, making members of a VTuber and their community an ideal target for our study.

To investigate the members of VTubers on Bilibili, we compile a dataset encompassing all live streaming records (2.7 million live records with 3.6 billion chat messages in total) of 4.9k VTubers, along with 21k other streamers on Bilibili, from June 2022 to August 2023. This dataset includes almost all streamers with a moderate fanbase on Bilibili, representing the largest dataset of VTubers available, to the best of our knowledge. Exploiting this dataset, we focus on the following research questions:

- **RQ1:** How do the behaviors of members differ from those of non-members? How do these patterns change before and after they become a member? We posit that understanding these behavioral variations can assist in identifying potential members and improving strategies for member promotion and retention.
- **RQ2:** How do the chat messages sent by members differ from non-members? How do these chat patterns change before and after they become members? Besides identifying potential members, analyzing these communication patterns can also offer insights into the unique culture of the community among members, potentially guiding efforts in better community building.

- **RQ3:** Leveraging the findings from RQ1 and RQ2, can we develop a tool that assists VTubers in identifying viewers likely to become members in the future? While numerous studies have explored recommending streams to viewers, the inverse — identifying valuable viewers for streamers — remains unexplored. By flipping the problem, this tool addresses a crucial gap, offering VTubers a strategic advantage in cultivating a loyal and engaged community.

By answering these RQs, our findings include:

(1) Members' activity escalates in the period leading up to their decision to purchase a membership, but (interestingly) decreases after purchase. Only 8.9% of members renew their membership in the following month, and we find that the likelihood to renew is correlated with the decrease in activity level ($\chi^2$ test p-values all less than $10^{-280}$). This underscores the necessity for VTubers to implement more effective retention strategies to maintain member interest beyond the initial period. (§4.2)

(2) Members' chat content exhibits higher alignment with the session's chat environment compared to non-members, indicating a stronger level of engagement and familiarity. This alignment increases before the viewer become a member. We find this is because members become adept at using the unique conventions and interactive styles of the specific VTuber, highlighting the crucial role of unique community culture in fostering community growth. (§5.1)

(3) Members are *more* likely to send toxic chat messages than non-members, with 12% having sent harassment and 6% having sent sexual content, while only 2% non-members have sent such messages. Our qualitative analysis finds that these messages are not necessarily malicious but rather a problematic form of cultural "overstepping" interactions. This emphasizes the need for better and more flexible moderation. (§5.2)

(4) Leveraging the above findings, we show that it is possible to help VTubers identify a small group of viewers (from their thousands of viewers) who are likely to be open to becoming a member. By focusing more attention on this smaller group, VTubers can increase the likelihood of converting potential future members into actual members, thereby enhancing fan community building and monetization. (§6)

## 2 BACKGROUND

### 2.1 A Primer on VTubers

***The VTuber Concept.*** VTubers originated in Japan and have rapidly gained popularity since their debut in 2016. Initially, VTubers focused on uploading videos to YouTube. However, with the rise of online live streaming, live streaming became their primary activity [7]. A VTuber is an animated virtual avatar that performs in live video streams or recorded videos. These avatars are often voiced by actors known as Nakanohitos in Japanese. Typically, VTubers use half-body 2D avatars created with tools like Live2D, which capture the actor's facial movements to animate the avatar's expressions. See Figure 7 in the Appendix for examples. Additional body movements can be triggered within these programs using commands from desktop computers. VTubers with access to full-body motion capture systems can perform using 3D avatars, allowing

for a wider range of motion. Like real-person streamers, VTubers often interact with their audience by reading and responding to chat messages during streams.

***VTubers in China.*** VTubers in China have experienced rapid growth since 2021, following the exit of Japanese agency-based VTubers from the Chinese market for various reasons [33]. Bilibili is the primary platform for VTuber livestreams in China. According to VTBs.moe (a website that indexes and tracks information about Vtubers in China), there are over 6,000 active Vtubers listed, with the most popular one boasting 4.5 million followers. Additionally, 35 indexed Vtubers have more than 1 million followers each.

### 2.2 Livestreaming on Bilibili

As one of the leading video streaming platforms in China, Bilibili hosts a diverse range of streamers besides VTubers, encompassing sports, esports, gaming, arts, and more. In addition to the standard features found on other livestreaming platforms, Bilibili provides two advanced features, membership and bullet chat.

***Membership.*** Bilibili offers a distinctive membership system, officially called *guards*. Users have the option to pay a monthly fee of ~$24 USD to become a member of a particular streamer, with more premium tiers available at ~$280 USD and ~$2800 USD. This system resembles the paid subscription model on Twitch but is more expensive and comes with additional privileges. These privileges include having the member's name displayed beside the live streaming screen, showing notifications upon entering the room, and enjoying more prominent visibility in chat interactions. Furthermore, streamers often invite their members to join private chat groups, assist in moderating live streaming sessions, and sometimes organize meetings offline. This creates a very different dynamic, compared to typical paid viewers on other platforms.

VTubers, or rather their fans, are the primary users of this feature on Bilibili. According to official rankings [1], at the time of writing, VTubers occupy all of the top 10 spots among streamers with the most members, and 39 out of the top 50 streamers are VTubers. The VTubers on Bilibili, in total, have succeeded in attracting 152,000 paying members, generating a monthly revenue of $3.5 million USD [48]. Consequently, this membership system offers a valuable framework for exploring the core audience of VTubers, the community among VTubers and their fans, and the monetization tactics of VTubers.

***Bullet Chat.*** Bullet chat (also known as danmaku in Japanese and danmu in Chinese) [18] is a comment system originally introduced by the Japanese website Niconico. It allows viewers to post their comments on the screen during livestreaming, where they appear as floating, moving text, as depicted in Figure 8 in Appendix. In comparison to traditional comment systems like those on YouTube or Twitch live streams, bullet chat offers a more engaging real-time interaction experience for users [18]. Due to the timely and straightforward nature of bullet chats, usually, a large number of comments are posted by viewers during livestreaming sessions.

***Gift & Superchat.*** Similar to other mainstream platforms, Bilibili also offers common monetization avenues. Viewers have the option to purchase virtual gifts and donate them to streamers. Additionally, there's a superchat (SC) system in place, allowing users to pay for sending a special message that gets pinned at the top of the chat

column for a certain period. These features provide streamers with alternative methods of generating revenue.

## 3 DATA COLLECTION METHODOLOGY

***Target Streamers.*** First, we compile a list of target streamers. To accomplish this, we rely on `VTBs.moe`, an indexing website for VTubers on Bilibili; and `danmakus.com`, a website dedicated to indexing and archiving popular livestreaming sessions on Bilibili. We include all streamers indexed by these two websites, and obtain a list of 4.9k VTubers and 21k other streamers, covering almost all streamers with a moderately large fanbase on Bilibili.

***Data Collection of Livestream Sessions.*** For each target streamer, we collect data for all livestreaming sessions from `2022-06-01` to `2023-09-01`. The data encompasses the metadata of the livestreaming sessions, such as the time, duration, title, area, *etc.* of each session. The data also encompasses all interactions from viewers during the livestreaming sessions. These interactions include entering the livestream, chatting, gifting, and paying for membership. We provide a full data schema description in Appendix A.2. In total, we obtain a dataset of 2.7 million live sessions, with 10.7 billion interaction records, covering 3.6 billion chat messages. We also note that part of our collected data comes from the `danmakus.com` website. See Appendix A.3 for ethical considerations.

***Identifying Members.*** Recall, our study focuses on members (*i.e.,* viewers who have paid for a membership). Thus, it is necessary to compile a list of members from our collected dataset. We therefore compile two separate lists of (*i*) members, and (*ii*) non-members (for comparison). We search our collected dataset for any membership purchase record from the period `2022-07-16` to `2023-07-16`. This start and end date ensures we have data for a sufficiently long period (45 days) before and after the membership. For each paid membership, we add it to the member list if it is the *first time* the user has paid for a membership for this VTuber. If it is not the first time, it will not be included in either the member list or the non-member list, as it would be impossible for us to study their before-membership period. Then, we add to the non-member list all other viewers in the same livestreaming session who have *never* paid for a membership during our measurement period. Note, a viewer can appear multiple times if they participate in different livestreaming sessions. In total, we obtain 608k membership records from 337k unique viewers.

Throughout the remainder of this paper, we refer to "member(s)", even if our analysis covers the time period before they had paid for their membership. We use "members pre-purchase" and "members post-purchase" to refer to members before and after the purchase of membership. We use ***the "membership-ed" VTuber*** to refer to the VTuber that the membership pertains to.

***Chat Messages Preprocessing.*** We later analyse chat messages. To facilitate this, we convert all chat messages into their text embeddings, to provide greater semantic insight into the content. For this, we use the `text2vec-base-chinese` embedding model, which has been reported to have high overall performance [4, 51].

We also inspect the toxicity in chat messages. To quantify this, we employ the OpenAI moderation API to label all messages. This tool allows us to obtain toxicity scores (0–1) across various categories, such as sexual content, harassment, hate speech, and violence, *etc.*

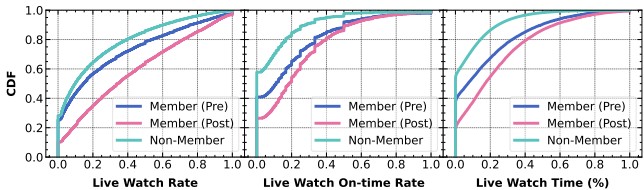

**Figure 1: The CDF of the viewing activity metrics to the membership-ed VTuber for members and non-members.**

We apply a threshold of 0.5 to determine whether a chat message is considered toxic in a given category.

## 4 CHARACTERIZING USER ACTIVITY

In this section we explore the activity patterns of members (**RQ1**). The online activity of a user is our first step in understanding their preferences, interests, and behavioral patterns [10, 54]. We posit that this insight can help VTubers understand how to identify potential future members effectively.

### 4.1 Comparing Members and Non-members

We begin by comparing the overall activity of members *vs.* non-members. Intuitively, we expect that members will exhibit higher levels of activity compared to non-members, as members are likely to be heavy users of the platform and more engaged with the membership-ed VTuber's live streaming.

***Methodology.*** We use several activity metrics (see Appendix A.4 for details) that have been widely used in previous studies of streaming platforms [14]. For each member and non-member, we compute each metric over a 90-day period. Specifically, for each user and the corresponding livestream session they join as a member, we consider a timeframe that spans 45 days *prior* to the livestream session (T-45) and 45 days *following* the session (T+45). This allows us to capture their behaviors before and after they become a member.

***Viewing Activity.*** We first analyze the viewing activity of members regarding the livestream sessions of the membership-ed VTuber. We evaluate this using three metrics: (*i*) the proportion of livestream sessions watched, (*ii*) the proportion of on-time attendance for livestream sessions, and (*iii*) the proportion of the watch time relative to the total streaming time of the VTuber.

Figure 1 presents the CDFs of the specified metrics over a 45-day period for members (pre & post purchase) and non-members. Note, for non-members, we combine the two 45-day periods as they are almost identical (this applies to all other subsequent figures in the paper unless otherwise specified). We see that members pre-purchase significantly outperform non-members in all three metrics. Specifically, members pre-purchase exhibit a higher live watch rate (mean 26% *vs.* 20%), a greater on-time rate (mean 18% *vs.* 9%), and an extended duration of watch time (mean 17% *vs.* 7%). This indicates that before they become a member, members begin to demonstrate a more dedicated viewing behavior, which provides insights into how to identify potential members.

***Bullet Chat Activity.*** We next investigate the bullet chat (see §2.1) activity of users — a more nuanced indicator of user engagement. A significant difference is observed both in the total number of chats sent across the platform, and in the average number of chats

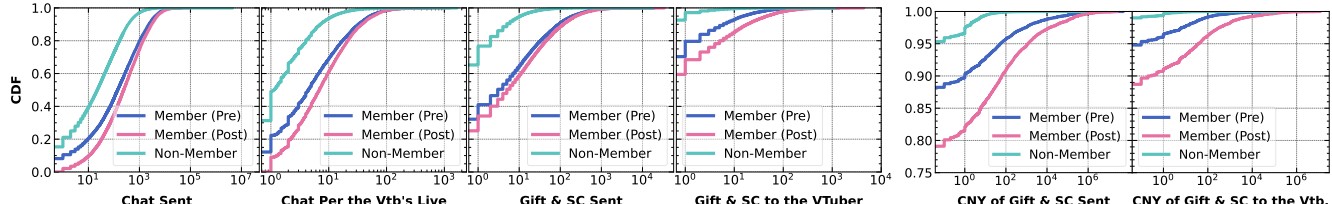

**Figure 2: The CDF of the metrics for chat and gift & superchat activity for members and non-members.**

sent per livestream session of the membership-ed VTuber. Figure 2 (a-b) displays the CDF for these two metrics. We see that members surpass non-members regarding the total number of chats sent, with averages of 876 for members pre-purchase *vs.* 161 non-members. A similar pattern is observed for the number of chats per the VTuber's livestream session, where members significantly outperform non-members, with an average of 12.9 for members pre-purchase *vs.* 3.2 non-members.

The findings indicate that members pre-purchase are indeed more engaged in terms of interaction with streamers. In contrast to viewing activity, this pattern is not limited to the membership-ed VTubers but also applies across the entire livestreaming platform. We argue that this serves as a valuable indicator for VTubers to select which viewers may be future members. We later exploit this for identifying potential members (§6).

***Gift & Superchat Activity.*** We finally delve into the gifts and superchats from viewers. Intuitively, considering users who have previously made purchases as potential members is a reasonable approach. Here, we evaluate this conjecture. Figure 2 (c-f) illustrates the CDF for both the quantity and monetary value of gifts and superchats sent by the user across the entire platform, and directed specifically towards the membership-ed VTuber.

It is evident that members consistently surpass non-members in all four metrics. In terms of platform-wide activity, the average for members pre-purchase are 48 (quantity) and 9190 (monetary value), significantly higher than 2.7 (quantity) and 2.6 (monetary value) for non-members. Furthermore, 12% of members pre-purchase send at least one gift or superchat, compared to only 4.5% of non-members.

Moreover, while the pattern of gift and superchat activity towards the membership-ed VTuber shows a similar trend to the platform-wide activity, a notable distinction exists. Specifically, 5% of members pre-purchase have previously sent a gift or superchat to the membership-ed VTuber, while 12% members pre-purchase have sent to any streamer. Curiously, this indicates that 7% of members pre-purchase have sent a gift but not specifically towards the membership-ed VTuber. This suggests that members pre-purchase are more engaged in terms of gifts and superchats, which is intuitive since it reflects a user's willingness to contribute (financially), even if the gifting may not be directed to the membership-ed VTuber.

Overall, these results confirm that considering users who have previously made purchases as potential members is a reasonable strategy to identify potential members. However, this strategy, while effective, is not sufficient on its own, given that there are still 88% of members who have not engaged in paid gifting or superchats before buying a membership. This highlights the complexity of identifying potential members, and underscores the need for a more nuanced understanding of user behavior and motivations.

### 4.2 Temporal Analysis

In §4.1, we find that engagement with the membership-ed VTuber significantly differs between members and non-members. We also observe that, among all metrics investigated (in Figure 1 and 2), members post-purchase exhibits higher values than members pre-purchase. Consequently, we are intrigued by how exactly the engagement evolves over time. We anticipate an upward trend as it approaches the time when they become a member. Thus, we next conduct a daily temporal analysis.

***Methodology.*** As the first step, we confirmed that the increase of the platform-wide activity is due to the heightened activity towards the membership-ed VTuber. Thus, we only consider the metrics in §4.1 regarding the membership-ed VTuber. However, instead of analyzing the 90-day period as a whole, we analyze it on a daily basis. For each individual day within the 90-day span, we aggregate the data for all members. For the first four metrics —- Live Watch Rate, Live Watch On-Time Rate, Live Watch Time, and Number of Chats Sent — we calculate the mean average. Meanwhile, for the metrics concerning the gifts and superchats sent, we aggregate them as the proportion of members who send at least one.

***Results.*** Figure 3 displays the time-series plots for the aggregated six metrics of member activity over the 90-day period. A consistent pattern emerges across all metrics: starting at its lowest point at T-45, the trend exhibits a steady increase, culminating in a peak at T+0. However, despite maintaining a relatively high level, a noticeable decline follows. Specifically, for the Live Watch Rate, Live Watch Time, and Number of Gifts Sent, the figures at T+45 drop to approximately the same levels as those observed at T-15.

***Explaining the Decline.*** The above observation confirms that member engagement intensifies during the lead-up to membership, but surprisingly declines soon afterwards. This raises intriguing questions about the factors that contribute to the observed decrease in engagement post-purchase. An intuitive explanation for the decline in engagement after membership could be a diminishing interest in the membership-ed VTuber. Thus, we investigate a key indicator for this: whether members renew their membership in the subsequent month. We note that due to the automatic subscription renewal mechanism, we might not capture this information if the VTuber does not stream when the renewal occurs. Thus, our analysis is confined to the instances where the VTuber is streaming in the following month when the renewal is expected to occur. This covers 67% of the membership subscriptions.

We find only 8.9% of members renew their membership in the following month. To further evidence this, we perform the $\chi^2$ test of independence [34] to examine the relationship between the decline in six engagement metrics and the membership renewal (See

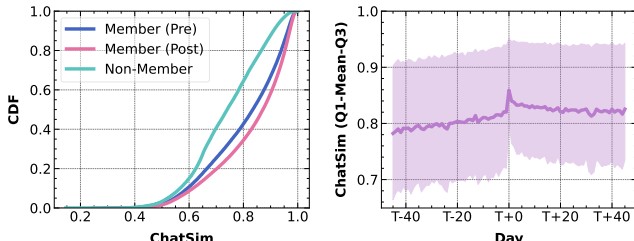

**Figure 3: The time-series plots of the activity metrics for members with the membership-ed VTuber.**

Appendix A.5 for details). We observe a clear correlation across all six metrics with significant p-values. This confirms that declining engagement is indeed related to lower renewal rates. Contrary to initial expectations, the findings suggest that purchasing a membership appears to be an *one-time* behavior. Viewers initially become increasingly engaged with the VTuber, leading them to subscribe. However, once they become members, roughly the first 2 weeks represent the peak of their activity. After this period, their interest seems to wane, as indicated by a decrease in engagement, with the majority choosing not to renew their membership.

This differs from previous research on Twitch subscriptions [50], which emphasizes the distinction between (continuous) subscriptions and one-time donations. We suspect that the high cost of membership (~$24 per month *vs.* $5.99 on Twitch) could be a contributing factor. Overall, the results underscore the need to reconsider membership strategies for VTubers, in stark contrast to prior membership systems. For instance, membership could be optimized by introducing shorter-term options, such as two-week memberships at lower prices. Additionally, this also highlights the necessity for VTubers to implement more effective retention strategies to maintain member interest after the initial period.

## 5 ANALYSIS OF LINGUISTIC BEHAVIORS

Given that members send more chats than non-member, with a noticeable increase leading up to their membership (§4), we next explore the messages sent by members (**RQ2**). We are particularly interested in two aspects. First, we wish to explore the presence of distinct community subcultures and community norms, in terms of how members interact with VTubers. Second, we wish to measure the potential harms that may be associated with this subculture. For example, concerns have been raised about the sexualization of VTuber avatars [49], and there are VTubers whose fan communities are notorious for engaging in online abuse and harassment [43, 55, 56]. Thus, we next explore these matters.

### 5.1 Similarity to the Chat Culture

We begin by examining the similarity between an individual viewer's chat messages and the overall chat environment (*i.e.,* culture) during a livestreaming session. Intuitively, a high level of similarity between a user and the remaining chat messages would suggest a robust engagement with both the livestreaming and the VTuber.

***Calculating the Similarity.*** To quantify this similarity, we introduce a metric termed **ChatSim** to capture the similarity between a viewer's chat messages and the overall chat environment in a livestreaming session. To compute ChatSim, we first generate embeddings for all chat messages using the methods outlined in §3. Subsequently, for each livestreaming session, we compute the average of all chat embeddings within that session to represent the

overall chat environment. Then, for each viewer in the session, we calculate the average of the embeddings for that viewer's chat messages. We then compute the cosine similarity between this average and the overall chat environment. This yields the ChatSim metric for the viewer in this livestreaming session.

***Comparing Members vs. Non-Members.*** We begin by comparing the ChatSim scores of members and non-members in livestreaming sessions. The analysis is conducted on a 90-day period that includes 45 days before the livestreaming session (T-45) and 45 days after the session (T+45), on users in our designated member and non-member lists, as described in Section 3. Figure 4a plots the CDFs.

It is evident that members exhibit higher ChatSim scores than non-members in the livestreaming sessions of the VTubers they support. The average (mean) scores are 0.80 for members pre-purchase, 0.83 for members post-purchase. This can be compared to just 0.74 for non-members. This suggests that members have higher alignment with the chat environment. We posit that this pattern could be helpful in identifying potential members.

Intuitively, this might be explained by the fact that the topic of chat discussions is closely related to the content the VTuber is streaming. However, we note that this is unlikely to result in a clear distinction between members and non-members. We therefore hypothesize that this is due to the community subculture among members and the specific norms how they interact with the VTuber, which we explore in the following paragraphs.

***Temporal Chat Analysis.*** In the previous paragraph, we noticed that ChatSim seems to rise after the purchase of the membership (Figure 4a). Thus, we next inspect how exactly ChatSim scores change before and after a user becomes a member for a VTuber. Figure 4b displays the ChatSim scores of members during the livestreaming sessions of the membership-ed VTuber, grouped by the number of days relative to the day they become members. We illustrate the average (mean) values and the first to third quartile range (Q1 to Q3) with shading.

**Figure 4: (a) The CDF of the ChatSim score for members and non-members in the livestream sessions of the membership-ed VTuber; (b) Daily average ChatSim score for members in the livestream sessions of the membership-ed VTuber.**

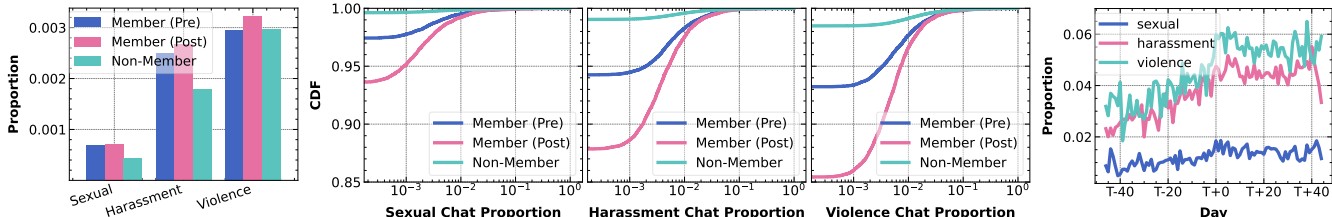

**Figure 5: (a) The proportion of toxic messages sent by members and non-members to the membership-ed VTuber; (b) The CDF of the proportion of toxic messages sent by members and non-members to the membership-ed VTuber; (c) The proportion of members who have sent a toxic chat messages to the membership-ed VTuber on the day.**

We observe an increasing trend before T+0, with the mean rising from 0.78 at T-45 to 0.82 at T-1, and reaching a peak of 0.86 at T+0. This trend is consistent with our findings on audience activity in §4.2. However, after T+0, there is a decline, with the mean decreasing to 0.83 at T+45. This slightly differs from the observations in §4.2: although a decrease is noted, it is minor and the mean remains higher than it was before T+0. We posit that this increasing pattern could be helpful in identifying potential members. However, the result (contrary to §4.2) suggests that the change of the ChatSim score is less relevant to the activity level of the member.

***Chat Topic Analysis.*** To gain a deeper understanding of the factors contributing to the differences (as compared to non-members) and the increases (over time) in ChatSim scores among members, we conduct a content analysis of chat messages. We use BerTopic [9] to extract the key topics from the chat messages. However, we find that the topics for each VTubers differ significantly. Thus, we build the topic model respectively for the 100 most popular VTubers's (ranked by the number of chats), and manually validate each one. The specifics of this analysis are detailed in Appendix A.6. Overall, our analysis shows that the most notably increased topics are associated with (*i*) endearments for the VTuber and the fans (increased for 100% of the selected VTubers), (*ii*) prescribed viewer reactions (98%), and (*iii*) slang/catchphrases/memes (95%). We emphasize that these are unique to each specific VTuber.

This indicates that members become familiar with the unique subculture and interactive styles of the VTuber. This aligns with findings from prior human-factors research on VTubers [30] and other streamers [35] which shows the importance of a dedicated viewer community with its own culture, and how this community and culture are established. This observation also helps explain why there is no significant decrease in ChatSim scores afterwards: although members may become less active over time, they still know the subulture and are able to interact as a "proficient" participant.

## 5.2 Toxicity

The presence of toxicity in chats, including sexual content, harassment, and violence, remains a contentious issue. On the one hand, VTubers may prefer to avoid such toxic chats as they could potentially harm the VTuber, affect the quality of the livestream, and negatively impact the audience experience. Yet, on the other hand, it can also be a consequence of high audience engagement in livestreaming. For instance, a VTuber who focuses on content that borders on erotic might not view chats containing sexual content as toxic and might even encourage such interactions [12, 46].

Similarly, livestreams dedicated to highly competitive esports often naturally attract comments that are violent in nature [19]. Thus, given the complex impact that toxic interactions can have on both a VTuber and their audiences, it is crucial to examine the role and extent of toxicity in chat messages to gain a better understanding.

***Methodology.*** To analyse the toxicity, we choose the 90-day period that includes 45 days before the livestreaming session (T-45) and 45 days after the session (T+45). Using the method described in §3, we label all chat messages in livestreaming sessions (within the 90-day period) related to the viewers included in our member and non-member lists (§3). Since we find minimal cases of other categories, we concentrate on three main toxicity categories: sexual, harassment (insulting or degrading someone), and violence among the categories of toxicity.

***Comparing Members vs. Non-Members.*** We first analyze the proportion of toxic messages. The results are presented in Figure 5a. Interestingly, the proportion of toxic messages from members pre-purchase is slightly *lower* than that of members post-purchase for sexual (0.68‰ vs. 0.71‰), harassment (2.5‰ vs. 2.7‰), and violence categories (2.9‰ vs. 3.2‰). While for non-members, the proportion for sexual (0.43‰), harassment (1.7‰), and violence (2.9‰) are all lower than members. This suggests that members are more inclined to send toxic chat messages, and that the frequency of toxic messages tends to increase after paying for membership.

Further, we compare the data at an individual viewer-level. Figure 5b shows the CDF of the proportion of toxic messages sent to the membership-ed VTuber out of all messages sent to the VTuber by each individual viewer. We see noticeable disparity across all three categories, with members significantly outpacing non-members, and members post-purchase significantly outpacing members-pre. Specifically, 6.4% members post-purchase sent sexual messages, 12.1% members-post sent harassment messages, and 14.5% members-post sent messages containing violence. Overall, the results confirm that members are more inclined to send toxic chat messages.

***Temporal Analysis.*** To further explore this, we next conduct additional temporal analysis on the proportion of members who have sent toxic chat messages to the membership-ed VTuber. Our previous analysis indicated a significant increase in this proportion after the membership purchase (as illustrated in Figure 5b). In Figure 5c, the results are presented on a daily basis. It is evident that all three categories exhibit a similar pattern to the ChatSim score shown in Figure 4b: a rise leading up to T+0, a peak at T+0, followed by a slight decrease. This alignment may be attributable to a similar

reason as the ChatSim score trend: toxic chat messages could potentially be a form of cultural engagement by some members, whereas they get more acquainted, they start to act differently.

**Content Analysis of Toxic Chats.** We therefore next inspect the precise content of the toxicity in chat messages, to clarify (*i*) how exactly this reflects viewer engagement; and (*ii*) whether such messages are culturally acceptable or require moderation. For the three categories, sexual, harassment, and violence, we respectively construct topic models within members' chat messages in the category, using BerTopic [9]. We present the details in Appendix A.7.

We find that sexual chats pose a significant issue. The top 10 topics (covering 94.6% of the sexual chats) predominantly revolve around sexual behaviors towards the VTuber (*e.g.,* "kiss" for 25.4%, "lick" for 24.3%) or direct references to sexual body parts (*e.g.,* "Hip" for 7,3%, "Chest" for 4.8%). It is unclear whether the VTuber is anticipating or merely tolerating it. Even if they are expecting such content, it may not be appropriate for a general purpose livestreaming platform and could offend other viewers.

Similar issues arise with harassment chat messages. The top 10 topics (covering 96.9% of harassment messages) all contain insulting language that is, by definition, harassment. However, many of these messages are characterized with a somewhat light-hearted tone (*e.g.,* "Stinky" for 5.1%, "Despicable" for 4%), blurring the line between problematic harassment and jests. Therefore, rather than a clear-cut need for moderation, as is the case with sexual chats, it may be left to the discretion of the VTuber to determine whether they can handle such comments to make the (potential) members pleased.

Violent chat messages appears to have two types. The first type is actually flirting that is similarly problematic, akin to sexual chat messages, *e.g.,* "Step-on/Trample" (23.3%). The second type appears broadly acceptable, which covers 8 out of the top 10 topics. These messages do contain violent language, such as "kill" (21.8%). However, these actions are typically directed towards a virtual entity (*e.g.,* in video games) or are actually memes.

The content analysis reveals that, as viewers become more familiar with the VTuber, this increased familiarity leads to more toxic behavior, including expressing their strong admiration for the VTuber through sexualized messages and making aggressive jokes that border on harassment. While such behavior may indicate high engagement, it also raises concerns about VTubers' online safety, particularly with the normalization of sexualized and aggressive interactions. Unlike previous research on harassment toward content creators [15, 44], this harassment notably originates from within the VTuber's own fanbase and paying supporters. This underscores the need for new mechanisms to address this complex issue, where platforms must develop and implement stronger and flexible moderation and support systems to protect VTubers from escalating harassment and ensure a safer online environment.

## 6 IDENTIFYING POTENTIAL MEMBERS

In this section, we propose a tool to assist VTubers in identifying potential members among their audience (**RQ3**). While prior studies have focused on recommending *streamers* to *viewers*, our approach contrasts by recommending *viewers* to *streamers*. As the level of attention and engagement provided by the streamer plays an important role in motivating viewers to make monetary contributions such as gifts or subscriptions [24, 31], we believe if a VTuber can pinpoint a specific group of potential members to focus on, it could increase the likelihood of converting them into real members. Leveraging the insights gained from **RQ1** and **RQ2**, we therefore design a tool, named FanRanker.[1]

### 6.1 FanRanker Design

To assist VTubers in focusing their member recruitment efforts, it would be useful to provide a ranking of the viewers watching the livestream, based on their likelihood of becoming a paid member. Thus, we develop a model that assigns a score from 0 to 1 to each viewers in the livestream. This estimates their probability of later becoming a member. Using these scores, FanRanker then generates a ranking of the top $n$ viewers most likely to become a member.

**Features.** Based on the insights from **RQ1** and **RQ2**, we select key features that exhibit significant disparities between members and non-members. To capture the temporal difference, we also incorporate the difference of the same feature across different time periods. For instance, consider a feature $F$, which we measure across two distinct time periods, yielding results $X_1$ and $X_2$. We then create a new feature, $F_\Delta$, defined as the difference $X_2 - X_1$. A comprehensive summary of these features is detailed in Appendix A.8.

**Model Training.** We experiment with five machine learning algorithms: Linear Regression (LiR), Logistic Regression (LoR), Random Forest (RF), Histogram-Based Gradient Boosting (HGB), and K-nearest Neighbors (KNN). To train the model, we utilize data in our member list and non-member list as described in Section 3. To prevent class imbalance, we undersample the non-member list. We employ 5-fold cross-validation and leverage grid search to tune the hyperparameters of each model. The hyperparameters used for each model are detailed in Appendix A.9.

### 6.2 FanRanker Evaluation

**Evaluation Metric.** To assess the effectiveness of FanRanker's ranking, we apply a ranking performance metric, which is defined by the position in the ranking of users who actually purchase a membership in this session. Specifically, for every membership paid by a viewer, $M$, during a live streaming session, $S$, we take a list $L_{other}$ of non-member viewers in $S$. We then apply the trained model to estimate the probability, $P_M^S$, that viewer $M$ will pay for a membership in session, $S$. Additionally, we calculate the probability, $P_O^S$, for each non-member viewer $O$ in $L_{other}$ for the same session, $S$. After obtaining these probabilities, we rank $P_M^S$ along with all $P_O^S$ values and determine the rank position of $P_M^S$ within this list. The ranking performance metric is defined as this rank position for the viewer $M$. A higher position indicates a more accurate ranking prediction, as it means the users who actually purchase a membership are ranked higher. Intuitively, other viewers with a high ranking (who have not yet purchased the membership) can be identified as "potential members", suggesting that VTubers should give more attention to them.

**Results.** We calculate the ranking performance metric and obtain results in both raw ranking position and percentile for each member (*i.e.,* top XX% among viewers in the livestreaming session). The

---

[1] Will be openly available in https://github.com/omitted-for-double-blind-review

results for the five models are illustrated as CDFs in Figure 6. An effective prediction would result in all future members attaining a high rank. We observe that both Random Forest and Histogram-Based Gradient Boosting exhibit similar performance, achieving the best outcomes, followed by K-nearest Neighbors, with linear regression and logistic regression trailing behind.

For the top-performing models (Random Forest and Histogram-Based Gradient Boosting), we find that 52% of viewers who purchase a membership during the livestreaming session are ranked in first place, while 85% fall within the top 50 positions. This indicates that, in the majority of cases, the model can accurately pinpoint viewers who are likely to subscribe within a small pool of users (top 50). Thus, our tool can successfully assist VTubers in identifying a manageable group of, say, 50 viewers, out of thousands. This smaller group of viewers then makes it practical for the VTuber to pay additional attention on, which may help convert more "potential" members into actual members. Overall, the results indicate that the tool is effective and can be be utilized in practical scenario if enough data is provided.

***Feature Importance.*** To gain a deeper understanding of the important features utilized in the models, we briefly analyze feature importance using the permutation feature importance method [2]. We focus on the two top-performing models (Random Forest and Histogram-Based Gradient Boosting). The ten most important features are showed in Figure 10 (Appendix), with the importance values scaled to a range of 0-1 using min-max scaling.

We observe that the most important features are linked to the ChatSim score (1st/2nd important for HGB and 2nd/4th important for RF) and the number of chats sent (3rd/4th important for HGB and 1st/3rd important for RF), along with other activity metrics with the specific VTuber. This is intuitive as chat activity serves as a direct measure of engagement, with the number of chats and ChatSim score indicating the quantity and quality of interactions, respectively. Additionally, the other activity metrics reflecting engagement with the VTuber also hold considerable importance in our models. We conjecture that many VTubers likely learn such patterns themselves, building an intuition for which viewers are most likely to subscribe. Our tool automates and simplifies this process, dramatically reducing the number of candidate viewers a VTuber needs to review.

## 7 RELATED WORK

***VTubers.*** The emergence of VTubers has influenced online communities and digital culture, drawing attention from various angles [7, 30]. Studies by Miranda et al. [32] and Rohrbacher & Mishra [38] explore the global expansion of VTubers, highlighting differences in cultural reception and integration, with a particular focus on the influence of VTubers in Portugal, the USA, and Austria. Research by Xu & Niu [52] delves into the psychological attributes of VTuber viewers, shedding light on the emotional and psychological factors that drive viewer engagement with VTubers. Furthermore, studies by Lee, Sebin & Lee, Jungjin [22] and Wan & Lu [49] examine the community and social aspects of VTubing, including its impact on fandom experience and how it offers new avenues for exploring and expressing identity, particularly in relation to gender dynamics and community building. Additionally, Turner [45] and Chinchilla & Kim [3] provide insights into the complex relationship between

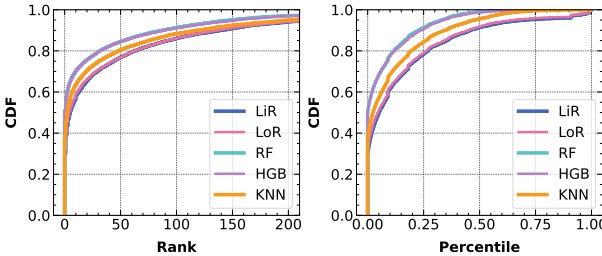

**Figure 6: The CDF of the rank and the percentile of the user who actually purchases a membership in the livestreaming session among other viewers in this livestreaming session.**

VTubers and identity, highlighting the role of VTubing in supporting marginalized communities and affecting perceptions of identity and social interaction.

While previous studies have relied on qualitative research methods such as interviews with VTubers or their viewers, our research represents the first data-driven investigation into VTubers' viewers using a large-scale dataset. Through this approach, we uncover previously unstudied characteristics of core VTuber viewers.

***Paid Subscriptions for Livestream.*** There are several livestream platforms offering a paid subscription or membership mechanism, and numerous studies have investigated this. Kobs et al. and Yu & Jia explore how users' interaction behaviors and donations through paid subscriptions contribute to the platform's vibrant community [21, 53]. Kim et al. and Wohn et al. investigate the potential to identify paying viewers through sentiment analysis of chat messages and examine the motivations behind digital patronage, respectively [20, 50]. Lee et al. employ multimodal analysis to deepen insights into subscription dynamics [23]. Work by Bründl et al. and Hilvert-Bruce et al. delve into synchronous participation and socio-motivational aspects of viewer engagement, highlighting the importance of active participation and community in influencing subscription behaviors [16, 40]. Houssard et al. further address monetization inequalities within Twitch, offering insights into the economic disparities faced by creators [17].

Our research differs in two key ways. First, while previous studies have concentrated on recommending streamers to viewers, our work shifts the focus towards assisting streamers in identifying potential members. Second, Bilibili's membership system distinguishes itself by being more expansive and offering additional features. We are the first to study the impact of these additional features on VTuber monetization.

## 8 CONCLUSION

This paper presented a comprehensive study of the core viewers of VTubers on Bilibili, the primary platform for VTuber livestreaming in China. Our findings offer valuable insights into the behaviors and characteristics of core VTuber viewers, which we use to develop a tool that can help VTubers identify potential high-quality viewers and effectively grow their fan communities. Additionally, our results underscore the challenges of retaining core viewers, building a unique fan community culture, and moderating toxic behaviors during livestreams. In the future, we aim to extend our analysis to other platforms, such as YouTube for Japanese VTubers and Twitch for non-Asian VTubers.

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

# A APPENDIX

## A.1 Sample Images of VTubers & Bullet Chats

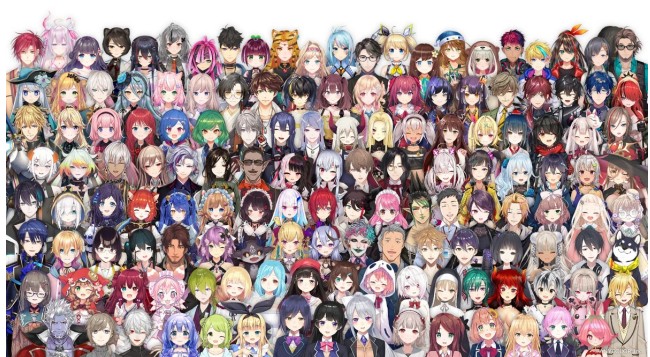

**Figure 7: VTubers (as in 2D avatars) affiliated with Nijisanji, a Japanese VTuber agency. The image is from fandom wiki. https://virtualyoutuber.fandom.com/wiki/NIJISANJI**

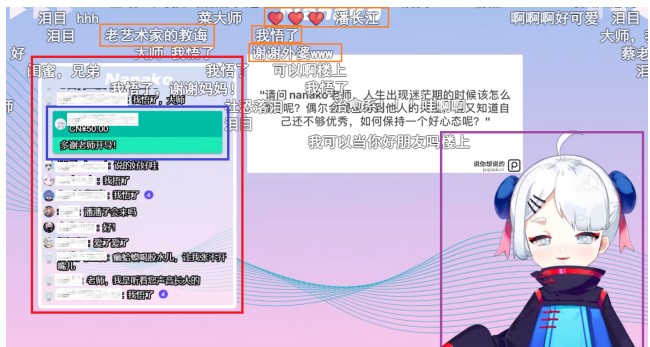

**Figure 8: Screenshot of a livestreaming session on Bilibili of 菜菜子Nanako (https://space.bilibili.com/595407557), a Chinese VTuber. The purple indicates the VTuber 2D avatar. The avatar is automatically animated, in-line with the human operator's movements. The orange indicates (some of) the bullet chats sent by the viewers. The red indicates the listing view of the bullet chats. The blue indicates the superchat.**

## A.2 Data Description

| Field | Type | Description |
|---|---|---|
| uId | Integer | Unique identifier for the streamer. |
| uName | String | Name of the streamer. |
| liveId | String | Unique identifier for the live session. |
| parentArea | String | The parent category or area of the live session. |
| area | String | The specific area or category of the live session. |
| coverUrl | String | URL of the cover image for the live session. |
| startDate | Integer (Epoch) | Start time of the live session, represented as a Unix timestamp. |
| stopDate | Integer (Epoch) | Stop time of the live session, represented as a Unix timestamp. |
| title | String | Title of the live session. |

**Table 1: Description of data fields for live session**

| Field | Type | Description |
|---|---|---|
| uId | Integer | Unique identifier for the user who sent the interaction. |
| uName | String | Name of the user who sent the interaction. |
| type | Integer | Type of the interaction. |
| sendDate | Integer (Epoch) | Time when the interaction was sent, represented as a Unix timestamp. |
| message | String | Content of the interaction message. |
| price | Integer | Price associated with the interaction, if any. |
| count | Integer | Count associated with the interaction, if any. |

**Table 2: Description of data fields for viewer interaction**

## A.3 Ethical Considerations

All data used in this paper is publicly available, and accessible to all Bilibili users. We collect bullet chats and other publicly displayed messages within live streaming sessions on the Bilibili platform. This information is intended to be openly visible for any viewer. We record the sender's anonymous ID, the content of the message, and the corresponding timestamp. It is important to note that we do not, nor are we able to, associate the user IDs on Bilibili with real personal identities. We do not analyze individual users but rather aggregate the data to gain a high-level understanding. We use danmakus.com to retrieve live streaming session data. The website states that the data can be used for research purposes, provided the source is acknowledged. We have IRB approval.

## A.4 Activity Metrics

***Viewing Activity Regarding the Membership-ed VTuber.*** We use the following metrics:

- **Rate of the VTuber's livestream sessions watched:** Number of the membership-ed VTuber's livestream sessions watched by the user, divided by the total number of the VTuber's livestream sessions.
- **Rate of the VTubers's livestream sessions watched on-time:** Number of the membership-ed VTuber's livestream sessions watched on-time by the user, divided by the number of the VTuber's livestream sessions watched by the user. If the user enters the livestream session within 10 minutes of its start, we classify this as "on-time".
- **Proportion of time of the VTuber's livestream sessions watched:** The total time that the user spends watching the membership-ed VTuber's livestream sessions divided by the total time of the VTuber's livestream sessions. We estimate the watch time for each session by subtracting the earliest timestamp of the user's activity in that session from the latest timestamp of their activity.

***Platform-wide Viewing Activity.*** We analyze the platform-wide viewing behavior of users by using the following three metrics:

- **Active days:** Number of active days of the user.
- **livestream sessions watched:** Number of livestream sessions watched by the user.
- **Streamers watched:** Number of streamers watched by the user.

Figure 9 presents the CDFs of the specified metrics over a 45-day period for both members (pre & post) and non-members. For non-members, we combine the two 45-day periods as they are almost identical. This practice also applies to all other subsequent figures in the paper unless otherwise specified. Contrary to our initial hypothesis, we see there is no significant difference in viewing activity between members and non-members.

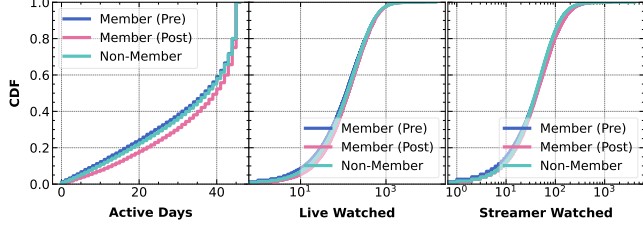

**Figure 9: The CDF of the metrics for platform-wide viewing activity of the members and non-members.**

***Bullet Chat Activity.*** We use the following measures:

- **Number of chats:** Total number of chats sent by the user.
- **Number of chats per the VTuber's livestream session:** The total number of chats sent by the user to the membership-ed VTuber's livestream sessions, divided by the number of VTuber's livestream sessions watched.

***Gift & Superchat Activity.*** We use the following measures:

- **Number of gifts and superchats:** Total number of gifts and superchats sent by the user

- **Value of gifts and superchats:** Total monetary value of gifts and superchats sent by the user.
- **Number of gifts and SCs to the VTuber:** Total number of gifts and SCs sent to the membership-ed VTuber by the user.
- **Value of gifts and SCs to the VTuber:** Total monetary value of gifts and SCs sent to the membership-ed VTuber by the user.

## A.5 Chi-square Test of Independence

We employ the Chi-square test of independence to investigate the relationship between the decline in six engagement metrics and membership renewal. The analysis utilizes the same metrics as detailed in Section 4.2. Specifically, we calculate the metrics for each 15-day interval, starting from 45, 30, and 15 days before the user became a member (T-45, T-30, and T-15), and then for the periods of 0, 15, and 30 days after the user became a member (T+0, T+15, and T+30).

To analyze the decrease, we subtract the metrics at T+30 from those at T+0. We then categorize the results into three groups: an increase is denoted by 1, a decrease by -1, and no change by 0. Finally, we calculate the Point-Biserial Correlation Coefficient to examine the correlation between the decrease in the six metrics and membership renewal, respectively. The results are presented in Table 3.

| Metric | Chi-square Statistic | p-value |
|---|---|---|
| Live Watch Rate | 1288.17 | $< 10^{-280}$ |
| On Time Rate | 3014.12 | $< 10^{-280}$ |
| Live Watch Time | 3916.37 | $< 10^{-280}$ |
| Chat per Live | 6892.15 | $< 10^{-280}$ |
| Gift & SC to the VTuber | 8090.92 | $< 10^{-280}$ |
| Gift & SC Value to the VTuber | 3627.77 | $< 10^{-280}$ |

**Table 3: Chi-square test of independence of the decrease of engagement metrics and membership renewal.**

## A.6 Topic Analysis of Members' Chat Messages

***Methodology.*** We employ BerTopic with text embeddings (embeddings delineated in Section 3). For each VTuber, we construct models of the topics present within members' chat messages over distinct 15-day intervals. These intervals commence at 45, 30, and 15 days preceding the date when a user joins as a member (denoted as T-45, T-30, and T-15, respectively), and extend to include the periods of 0, 15, and 30 days subsequent to membership commencement (denoted as T+0, T+15, and T+30, respectively). We restrict the topic count to a maximum of twenty.

***Findings.*** We discern an uptrend in the prevalence of five specific categories of topics across the most of the top 100 VTubers.

- **Endearments for the VTuber (100% of the investigated VTubers)**: Fans invariably use a term of endearment when referring to the VTuber.
- **Self-referential Terms (100% of the investigated VTubers)**: Similarly, there exists a specific term that the VTuber uses to address their fans, which, in turn, becomes the term by which the fans identify themselves.

- **Prescribed Viewer Reactions (98% of the investigated VTubers)**: There are established conventions dictating how viewers should react to certain events about the VTuber. For example, these conventions may apply when the VTuber wins or loses a game, or engages in humorous or silly behavior, etc.
- **Slang-Associated Entities (95% of the investigated VTubers)**: This category encompasses entities closely tied to the VTuber, which can include another streamer, real-life or virtual friends, family members, pets, or places associated with the VTuber, etc. These entities often have various slang terms linked to them.
- **Abbreviations Using Pinyin Initials (88% of the investigated VTubers)**: Often, abbreviations are used that consist solely of the initial letters in Pinyin. Regrettably, in the majority of cases, we are unable to decipher these abbreviations. However, we hypothesize that they typically pertain to one of the aforementioned four categories.

## A.7 Topic Analysis of Toxic Chat Messages

***Methodology.*** We employ BerTopic with text embeddings (embeddings delineated in §3). For the three categories, sexual, harassment, and violence, we construct models of the topics present within members' chat messages in the category. We then use the embeddings strategy[2] to reduce the outliers.

We list the top 10 topics for sexual, harassment, and violence chat messages in Tables 6, 7, and 8. **Caution: For transparency, we present each topic as it is. Therefore, the tables may contain words that some may find disturbing or offensive.** In each category, the top 10 topics encompass the 94.6%, 96.9%, and 85.3% of the chat messages, respectively.

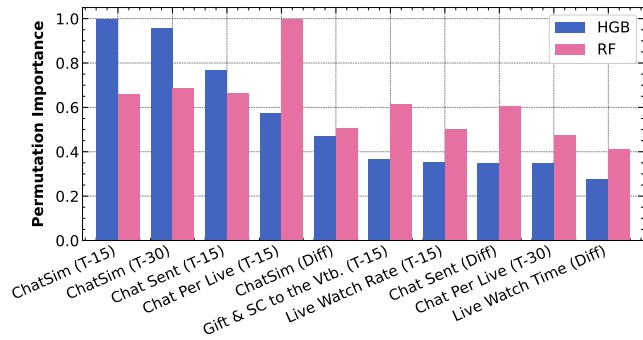

**Figure 10: Permutation feature importance for Histogram-based Gradient Boosting and Random Forest, with importance values scaled to a range of 0-1 using min-max scaling.**

---

[2]https://maartengr.github.io/BERTopic/getting_started/outlier_reduction/outlier_reduction.html#embeddings

## A.8 Features

| Feature | Ref. | Measured | Diff. |
|---|---|---|---|
| Chat Sent | §4.1 | T-30 & T-15[1] | None |
| Gift & SC | §4.1 | T-30 & T-15 | None |
| Gift & SC (in CNY) | §4.1 | T-30 & T-15 | None |
| Live Watch Rate | §4.1 | T-30 & T-15 | T-45 *vs.* T-15[2] |
| Live Watch On-time Rate | §4.1 | T-30 & T-15 | T-45 *vs.* T-15 |
| Live Watch Time | §4.1 | T-30 & T-15 | T-45 *vs.* T-15 |
| Chat Per Live | §4.1 | T-30 & T-15 | T-45 *vs.* T-15 |
| Gift & SC to the Vtb. | §4.1 | T-30 & T-15 | T-45 *vs.* T-15 |
| Gift & SC (in CNY) to the Vtb. | §4.1 | T-30 & T-15 | T-45 *vs.* T-15 |
| ChatSim (Average)[3] | §5.1 | T-30 & T-15 | T-45 *vs.* T-15 |
| Sexual (Binary)[4] | §5.2 | T-30 & T-15 | None |
| Harassment (Binary) | §5.2 | T-30 & T-15 | None |
| Violence (Binary) | §5.2 | T-30 & T-15 | None |

1. Result measured {from T-30 to T-15} and result measured {from T-15 to T-0}. The current livestreaming session is not included.
2. Calculated by subtracting the result measured {from T-15 to T-0} from the result measured {from T-45 to T-30}.
3. Calculated by taking the average ChatSim for all live streaming sessions of the VTuber throughout the measurement period.
4. A boolean value that is True if the user send such chats to the VTuber during the measurement period.

**Table 4: Features used for machine learning models.**

## A.9 Model Hyperparameters

| Algorithm | Parameters |
|---|---|
| LiR | - |
| LoR | penalty=L2, C=1.0 |
| RF | n_estimators=200, max_depth=16 |
| HGB | learning_rate=0.1, max_iter=100, max_depth=None |
| KNN | n_neighbors=50, leaf_size=30, p=2, metric=minkowski |

**Table 5: Hyperparameters Used for Each Model.**

## A.10 Feature Importance

Figure 10 presents the feature importance of the two top performing FanRanker models: Random Forest and Histogram Based Gradient Boosting.

| | % | Keyword | Explanation |
|---|---|---|---|
| Rank | | | |
| 1 | 25.4% | 亲 (Kiss) | - |
| 2 | 24.3% | 舔 (Lick) | In a sexual context |
| 3 | 10.8% | 屁 (Fart, Butt) | - |
| 4 | 8.2% | 舔、摸 (Lick, Touch) | In a sexual context |
| 5 | 7.8% | 汁 (Juice) | In a sexual context |
| 6 | 7.3% | 臀、屁股 (Hip, Butt) | - |
| 7 | 4.8% | 熊、奈子 (Breast) | Slangs |
| 8 | 2.6% | 仙女棒 (Vibrator) | Slangs |
| 9 | 2.0% | 裸 (Naked) | - |
| 10 | 1.4% | 妈 (Mommy) | - |

**Table 6: Topics for toxic chat messages in the sexual category.**

| | % | Keyword | Explanation |
|---|---|---|---|
| Rank | | | |
| 1 | 40.8% | 可恶、你小子 (Bastard) | More likely to be playful or teasing instead of Harassment |
| 2 | 21.6% | 笨 (Stupid) | Traditionally considered insulting, but it can also be playful or teasing with less hostility, depending on the context |
| 3 | 16.3% | 傻 (Stupid) | Insulting, but its hostility has diminished over time, unless used in combination with more aggressive words |
| 4 | 5.1% | 臭 (Stinky) | More likely to be playful or teasing instead of Harassment |
| 5 | 4.5% | 坏女人 (Bad Woman) | Traditionally considered insulting, but it can also be playful or teasing with less hostility, depending on the context |
| 6 | 4.0% | 渣、卑鄙 (Despicable) | Traditionally considered insulting, but it can also be playful or teasing with less hostility, depending on the context |
| 7 | 1.9% | Animal Related | Many VTubers' avatars have the design elements from animals |
| 8 | 1.1% | 秃 (Bald) | - |
| 9 | 1.0% | 讨厌你 (Hate you) | - |
| 10 | 0.6% | 小丑 (Clown) | The insult emphasizes a lack of respect for the person's abilities or behavior, reducing them to a figure of mockery or ridicule |

**Table 7: Topics for toxic chat messages in the harassment category.**

| | % | Keyword | Explanation |
|---|---|---|---|
| Rank | | | |
| 1 | 23.3% | 踩 (Step-on, Trample) | In a sexual context |
| 2 | 18.3% | 吃我一拳 (Take my punch) | A meme [39], usually with no actual violent meaning |
| 3 | 15.5% | 杀 ... (Kill something) | Kill something, typically in a gaming context |
| 4 | 8.1% | Knife Related | Typically in a gaming context |
| 5 | 6.3% | 杀 (Kill) | Kill, while without a direct object, typically to express emotion or give instructions in a gaming context |
| 6 | 4.7% | 吞 (Swallow) | Typically used with slangs with no violent meaning |
| 7 | 2.6% | 猎杀 (Hunt) | Typically in a gaming context |
| 8 | 2.0% | Gun Related | Typically in a gaming context |
| 9 | 2.0% | 毁灭 (Destroy) | Typically used to express emotion in a exaggerated way |
| 10 | 1.5% | 炸 (Explode) | Typically used to express emotion in a exaggerated way |

**Table 8: Topics for toxic chat messages in the violence category.**

