# OpenReview forum: "Virtual Stars, Real Fans: Understanding the VTuber Ecosystem"
_ACM.org/TheWebConf/2025/Conference — WWW 2025 Poster_

### Official Review · Reviewer_RLeJ · 2024-11-23

**Novelty:** 5
**Technical Quality:** 4

**Review:**

This paper explores the behaviors and engagement patterns of VTuber viewers on the Chinese livestreaming platform Bilibili, analyzing a large dataset of 2.7 million livestream sessions. It identifies key characteristics of members, including their increased activity pre-membership, alignment with chat culture, and propensity for toxic interactions, while highlighting challenges such as membership retention and moderation. The study introduces "FanRanker," a tool designed to help VTubers identify potential future members, reversing traditional viewer-recommendation systems to enhance community-building and monetization.

Please refer to the strengths and weaknesses for more details.

**Strengths**

1. The paper investigates an interesting novel problem; shifting the focus from recommending content to viewers to recommending viewers to streamers. This is a creative application of recommendation systems to a new context and fills an unexplored niche within livestreaming research.
2. The dataset used in this study is the largest and most detailed to date for VTuber research, including 2.7 million livestream sessions, 3.6 billion chat messages, and 608,000 membership records. The dataset's granularity enables a detailed examination of viewer behavior, making the findings robust. I would think that the dataset is an important contribution for this study. However, there authors have made no signs of publishing this dataset, limiting its contribution and reproducibility.

**Weaknesses**
1. The technical novelty of this paper is rather limited. While the reverse recommendation approach is conceptually novel, the underlying techniques (e.g., Random Forest, Gradient Boosting) and behavioral metrics are well-established, resulting in limited methodological innovation. ChatSim, while innovative, is not rigorously validated through ablation studies or alternative engagement metrics. This limits the understanding of its true contribution to the tool's performance.
2. The lack of comparison with established recommendation systems (e.g., those used on Twitch or YouTube) weakens the empirical claims about the superiority or distinctiveness of FanRanker.
3. The evaluation relies predominantly on ranking performance without reporting additional metrics such as precision, recall, or F1-score. Statistical significance testing is also absent, which weakens the robustness of the results

**Questions:**

1. Did you consider comparing FanRanker with simpler or established baseline approaches, such as recommendations based on historical gift/superchat activity, chat frequency, or existing viewer engagement models? If not, why were they omitted, and could they be included for fairer evaluation?
2. How critical is ChatSim to FanRanker’s performance compared to other features? Can you provide evidence, such as an ablation study, to show its unique contribution?
3. Could you clarify how FanRanker is intended to be used by VTubers in practice? For instance, how might a VTuber interpret and act on the ranked recommendations? Have you received any feedback from VTubers about the tool’s usability or utility?

**Reviewer Confidence:**

4: The reviewer is certain that the evaluation is correct and very familiar with the relevant literature

**Scope:**

4: The work is relevant to the Web and to the track, and is of broad interest to the community

---

### Official Review · Reviewer_QQse · 2024-12-02

**Novelty:** 5
**Technical Quality:** 5

**Review:**

This paper is an analysis of the VTuber (virtual content creators) member community with a focus on people who prefer to watch livestreams of VTubers on Bilibili – a Chinese video-hosting/streaming platform. The authors filter VTubers live streaming sessions to explore user engagement, finding that people who are about to subscribe start engaging more with the streamers in terms of viewing, chatting, and other engagement metrics. They also uncover the subculture of the communities formed by the VTuber audience by performing linguistic analysis of the viewer live chats. Lastly, they built a classifier, FanRanker, that helps identify potential subscribers for the VTuber/streamer.

This paper correctly identifies an explorable niche that is gaining traction and requires a better understanding of virtual streaming. The writing and flow of the paper is good in how it helps the reader get familiar with the VTuber landscape. Overall, I would like to accept the paper with some changes.

One thing that was lacking was an explicit comparison of your findings (VTuber streaming) with non-virtual streaming on the same or similar platforms. It makes sense to compare them because if there is no difference between normal streaming audiences in VTubers then maybe we can use the same strategies to attract and retain a greater audience as other streamers (I would say those strategies have been well explored in the past).

Something that is strongly needed is a distribution of the VTuber audience. I am assuming that the 4.9k VTubers will have a long tail in terms of subscribers or members that follow them. In such a case, the results will be depictive of the top 100-200 streamers and their audience (an over representation). Do the top 20% of streamers follow the same pattern as the remaining 80%? What implications does the long tail have on the generalizability of the results?

The conclusive paragraphs of the results of some sections felt weak. They lack an implication of the effortful analysis that precedes it. For instance, considering the last paragraph of section 4.2, you postulate that the high subscription cost of Bilibili might be the reason why people don’t resubscribe. So, a new shorter-term subscription option to cater for VTubers. How is this subscription model exclusively catering for VTubers? This new subscription model is a platform wide policy catering equally for VTubers and non-VTubers. Do they also have a drastic decline in subscriptions after a month?

The idea of developing a classifier for streamers feels unnecessary. A person who engages more with you is likely to subscribe, so you cater more to them is the crux of the classifier. Do we need it? There was no motivation for it. I suggest the space can be better used for further developing the original story by showing VTuber audience and engagement distributions and then comparing them with non-VTubers.

**Questions:**

Is the analysis just on the VTubers data from the dataset or does it include all the streamers’ data? The authors should explicitly state what data they used. Further, the repetitive mentioning of 21k other streamers without using them adds no value to the paper;on the contrary, it adds to the readers confusion.

**Reviewer Confidence:**

3: The reviewer is confident but not certain that the evaluation is correct

**Scope:**

4: The work is relevant to the Web and to the track, and is of broad interest to the community

---

### Official Review · Reviewer_Qtzb · 2024-12-02

**Novelty:** 6
**Technical Quality:** 4

**Review:**

The paper investigated the VTuber ecosystem from the perspective of viewers and conducted quantitative analysis on large-scale data. It explores the underexplored area of user behaviors such as engagement, retention, and toxic messages, providing insights for future research. The author also introduced "FanRanker" as a tool to recommend potential members to VTubers, which is different from a traditional recommending system.

The study explored an understudied topic and the quantitative study is well-conducted. However, I think the depth of the paper can be improved through:
1. more elaboration on the methodology, such as how the original Chinese bullet chats are translated to be analyzed with OPENAI moderation API and how the activity metrics are derived from previous research (see the questions section for details)
2. comprehensive evaluation of the FanRanker. Currently, the authors only provide a statistical evaluation of the tool. The utility of the tool would be more convincing if stakeholders' feedback or some synthesis with the existing work about VTuber's needs about recommending tools could be included
3. deeper discussion beyond the presentation of numbers. Though the statistical results are convincing, I found the paper lacks a deeper discussion regarding actionable insights. For example, the authors argued that the toxicity analysis indicates a need for stronger and flexible moderation and support from the platforms. Some concrete design implications built upon the current platform moderation policies would be helpful.

**Questions:**

1. I assume the chat history you collected is in Chinese, can you elaborate on how you dealt with the translation before using the moderation API to classify toxicity? Also, a detailed exploration of the platform's moderation policies and their impact on community dynamics would add depth.
2. In lines 319-320 (Section 4.1), can authors be more explicit on how the activity metrics (appendix A.4) are derived from previous research, especially from reference 14? I read reference 14 and am confused about the rationale of the usage of the activity metrics in Appendix A.4, it seems like the paper didn’t mention any specific metrics.

**Reviewer Confidence:**

3: The reviewer is confident but not certain that the evaluation is correct

**Scope:**

4: The work is relevant to the Web and to the track, and is of broad interest to the community

---

### Official Review · Reviewer_Xs4M · 2024-12-02

**Novelty:** 4
**Technical Quality:** 4

**Review:**

The paper is methodologically robust, relying on a large-scale dataset (2.7 million live sessions and 3.6 billion chat messages) to derive insights. It demonstrates a thorough understanding of VTuber ecosystems and applies rigorous machine learning models (e.g., Random Forest, Histogram-Based Gradient Boosting) to develop a practical tool, FanRanker. The analysis covers multiple dimensions, including user activity, linguistic behavior, and toxicity in chats, showcasing the authors' ability to extract meaningful conclusions from extensive data.
The structure of the paper is clear and logical. Each research question (RQ) is explicitly stated and addressed systematically in its own section. Figures and tables complement the text well, providing a clear visual representation of the findings.

The study is somwhat platform-specific, which limits its generalizability. Although the authors plan to extend their work to other platforms, this is not addressed in sufficient depth within the current paper.
While the concept of recommending viewers is unique, some aspects, such as analyzing user activity and engagement, build on existing work. A more explicit comparison with prior studies would help establish the originality of this approach.
Some figures lack clear legends or context in their captions (e.g., Figure 3 on temporal analysis of engagement metrics)

**Questions:**

The paper discusses general audience engagement patterns but does not account for the specific personality or content style of individual VTubers, which could influence viewer behavior. For example, some VTubers may foster more intimate or aggressive fan cultures, which could impact both the level of engagement and the types of toxic behavior seen in chats.If the authors could discuss how VTuber personalities or content themes might vary and influence the results, this would help provide a clearer understanding of the diversity in fan behaviors.

**Ethics Review Description:**

The paper touches briefly on data collection ethics but does not sufficiently discuss concerns regarding user privacy

**Ethics Review Flag:**

Yes

**Reviewer Confidence:**

3: The reviewer is confident but not certain that the evaluation is correct

**Scope:**

4: The work is relevant to the Web and to the track, and is of broad interest to the community